# Enteroaggregative *Escherichia coli* in mid-Norway: A prospective, case control study

**Ingvild Haugan**[1,2]*, **Marit Gudrun Husby**[1], **Bjørg Skjøtskift**[1], **Dorothea Aamnes Mostue**[1], **Andreas Brun**[1], **Lene Christin Olsen**[1,2], **Melanie Rae Simpson**[3,4], **Heidi Lange**[5], **Jan Egil Afset**[1,2]

1 Department of Medical Microbiology, Clinic of Laboratory Medicine, St Olavs Hospital, Trondheim University Hospital, Trondheim, Norway, 2 Department of Clinical and Molecular Medicine, Faculty of Medicine and Health Sciences, Norwegian University of Science and Technology, Trondheim, Norway, 3 Department of Public Health and Nursing, Faculty of Medicine and Health Sciences, Norwegian University of Science and Technology, Trondheim, Norway, 4 Clinical Research Unit Central Norway, St Olavs Hospital, Trondheim University Hospital, Trondheim, Norway, 5 Norwegian Institute of Public Health, Oslo, Norway

* ingvild.haugan@stolav.no

## Abstract

### Background

The use of molecular methods has led to increased detection of Enteroaggregative *Escherichia coli* (EAEC) in faecal samples. Studies have yielded conflicting results regarding the clinical relevance of this finding. The objective of this study was to investigate the prevalence of EAEC in faecal samples from patients with diarrhoea and healthy controls and describe characteristics of EAEC positive persons.

### Methods

From March 1$^{st}$, 2017 to February 28$^{th}$, 2019, we investigated all consecutive faecal samples from patients with diarrhoea received at the laboratory and collected faecal samples from randomly invited healthy controls from mid-Norway. Real-time multiplex PCR was used for detection of bacterial, viral, and parasitic pathogens. We registered sex, age, urban versus non-urban residency, and travel history for all participants. Statistical analyses were performed with Pearson chi-squared test, Kruskal-Wallis test, and Mann-Whitney U test.

### Results

We identified EAEC in 440 of 9487 (4.6%) patients with diarrhoea and 8 of 375 (2.2%) healthy controls. The EAEC prevalence was 19.1% among those with diarrhoea and recent foreign travel and 2.2% in those without travel history independent of diarrhoea. Concomitant pathogens were detected in 64.3% of EAEC-positive patients with diarrhoea. The median age was 28.5 in those with EAEC-positive diarrhoea and 38 in those with EAEC-negative diarrhoea (p <0.01). In patients with diarrhoea, travel was reported in 72% of those with EAEC and concomitant pathogens, and 54% and 12% in those with only EAEC and no EAEC, respectively (p <0.01).

**Data Availability Statement:** All relevant data are within the manuscript and its Supporting Information files.

**Funding:** The funder provided support in the form of salaries for authors IH, MGH, BS, DAM, AB, LCO, and JEA but did not have any additional role in the study design, data collection and analysis, decision to publish, or preparation of the manuscript. The specific roles of these authors are articulated in the 'author contributions' section.

**Competing interests:** The authors have declared that no competing interests exist.

## Conclusions

EAEC was a common detection, particularly in patients with diarrhoea and recent international travel, and was found together with other intestinal pathogens in the majority of cases. Our results suggest that domestically acquired EAEC is not associated with diarrhoea. Patients with EAEC-positive diarrhoea and concomitant pathogens were young and often reported recent travel history compared to other patients with diarrhoea.

## Introduction

Diarrhoeal diseases cause significant morbidity and mortality among children in developing countries [1, 2]. The World Health Organisation (WHO) estimated diarrhoea to be the second largest communicable cause of death globally and the eighth most common cause of total deaths in low-income countries in 2019 [3]. In high-income countries, acute diarrhoea rarely causes death, but still comes with a financial burden and may be linked to complications affecting both the gastrointestinal tract and other organs [4–7].

Diarrhoeagenic *Escherichia coli* (DEC) is a group of bacterial intestinal pathogens detected in cases of diarrhoeal diseases. Genetic and phenotypic profiles of pathogenicity define the DEC pathotypes, which include enteroaggregative (EAEC), enterotoxigenic (ETEC), Shiga toxin-producing/enterohaemorrhagic (STEC/EHEC), enteropathogenic (EPEC), enteroinvasive (EIEC), and diffusely adherent (DAEC) *Escherichia coli* [8].

Nataro *et al.* first defined EAEC in 1987 by how the bacteria aggregate and adhere to HEp-2 cells in a stacked-brick pattern [9], and this phenotypic feature is still the gold standard for identification of EAEC. In addition, EAEC must lack the ETEC toxins heat-stable toxin (ST) and heat-labile toxin (LT). EAEC has a high degree of heterogeneity [10–12], and a genotypic definition of EAEC does not yet exist [10]. Identification based on adherence in cell culture is labour intensive, and several genetic targets have been evaluated as possible markers of EAEC infection [13]. AggR is a regulator of multiple virulence factors and activates the expression of aggregative adherence fimbriae (AAF) which mediate EAEC adherence [14]. Only EAEC strains harbouring the *aggR* gene has been evaluated as a cause of diarrhoea in a volunteer study, and the clinical significance of *aggR* negative EAEC is more uncertain [10, 15]. Some authors have used the terms typical EAEC and atypical EAEC on *aggR* positive and *aggR* negative EAEC, respectively [16–18].

The heterogenic nature of EAEC and the use of different detection methods might largely explain why studies have yielded conflicting results regarding the association between EAEC and diarrhoea. Bacterial culture remains a common screening method for the detection of enteropathogenic bacteria in studies on the aetiology of infectious diarrhoea [19–21]. Subsequent identification of *E. coli* strains as EAEC requires a molecular based method and is not performed in all studies of bacterial gastroenteritis. The detection of EAEC is likely to increase if, instead of bacterial culture, a PCR method is applied as the initial diagnostic test directly on the stool sample, as shown in a study by Ochieng et al., where the detection rate increased by nearly fourfold [22].

The first described association between EAEC and illness was in childhood diarrhoea in Chile and was reported in the same publication that described the pathotype in 1987 [9]. Subsequent clinical studies have reported an association between EAEC and diarrhoea [23–27], persistent diarrhoea [28, 29] and growth impairment in children in developing countries [30–32]. Other studies have failed to find an association between EAEC and diarrhoea in children

[33, 34] or adults residing in industrial regions [35, 36]. The best-described association between EAEC and diarrhoea in high-income countries is travellers' diarrhoea [27, 37–43].

EAEC has been reported to cause large outbreaks of diarrhoeal disease, including the European outbreak of an EAEC/EHEC hybrid pathotype causing 782 cases of haemolytic uraemic syndrome (HUS) and 46 deaths [44–46], which affirms the pathogenic potential of at least some EAEC strains.

The carrier rate in the healthy population and the true disease burden of EAEC remain unknown in most regions of the world. EAEC was detected in 2% of 798 German children with diarrhoea compared to none in the healthy control group of 580 children [47], and in 4% of 719 faecal samples from 179 Danish day care children, out of whom 24% reported diarrhoea [48]. A recent Danish study found EAEC in 488 of 18,610 (2.6%) faecal samples when analysing all samples received at a microbiological laboratory over a period of one year with a multiplex PCR panel [49]. To our knowledge, this is the only study published so far that presents data from such syndromic panel testing of all samples from patients with diarrhoea received at a laboratory for a period of at least one year, and we believe there are no other published epidemiological studies on EAEC that include the adult population in any Nordic country.

In this study we aimed to 1) describe characteristics of patients presenting to a general practitioner or health care facility in Sør-Trøndelag County due to diarrhoeal disease from March 1st, 2017 through February 28th 2019 compared to a healthy control group, 2) examine their faecal samples for the presence of intestinal pathogens, and 3) describe the characteristics of patients with EAEC-positive diarrhoea and investigate the possible association between EAEC and diarrhoea.

## Methods

### Study setting

The study was conducted at the Department of Medical Microbiology, St. Olavs Hospital, Trondheim University Hospital, Norway. The department serves St. Olavs Hospital and most general practitioners and private practices in the former Sør-Trøndelag County, with approximately 320,000 inhabitants. The department is also the referral laboratory for hospitals and practitioners in the former Nord-Trøndelag County and the northern part of Møre og Romsdal County, with a combined population of approximately 200,000.

### Study populations

During the time period from March 1st, 2017 through February 28th, 2019, 11,386 faecal samples were received at the laboratory for analysis of gastrointestinal pathogens (Fig 1). Samples examined with the laboratory's gastrointestinal multiplex PCR panels were considered eligible for the project. We excluded samples that were only investigated with the parasitic panel and samples with other indications than gastro-intestinal disease specified on the request form. To avoid samples with post-infection shedding of bacterial DNA, we excluded multiple samples from the same person if they were taken within three months unless they revealed a different pathogen from the first sample. We assumed that samples from the same person taken further apart than three months represented new diarrhoeal episodes. Based on these criteria, faecal samples from 9,487 diarrhoeal episodes from 9,189 patients of all age groups were included in the study. The need for consent from this study group was waived by the ethics committee.

For the healthy control group, we invited approximately 4,400 people with permanent residency in Sør-Trøndelag County by random selection from the National Population Register throughout the study period. Invitation letters and written consent forms were posted from March 1st, 2017 through February 28th, 2019. For minors, written consent forms were

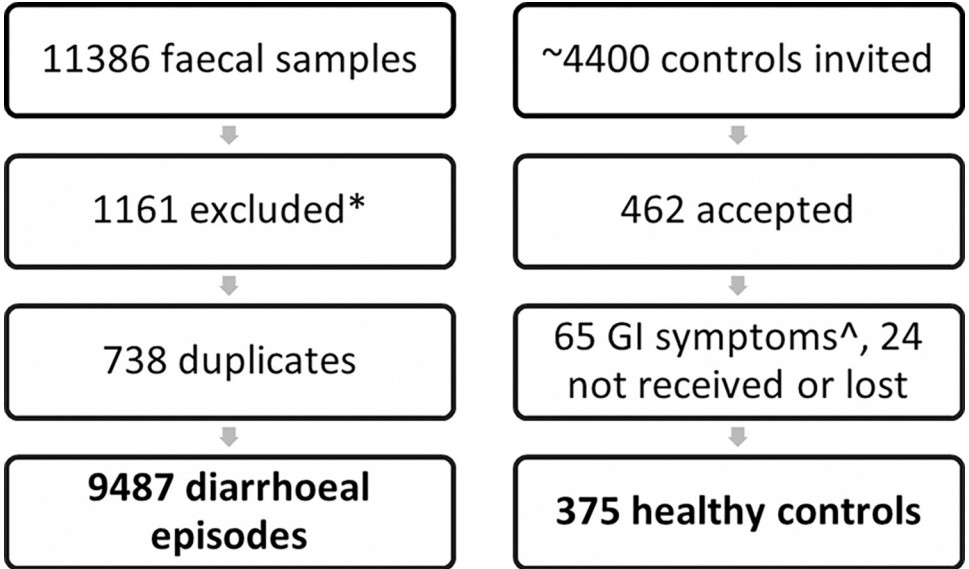

**Fig 1. Flowchart showing the inclusion process of patients with diarrhoea and healthy controls.** *781 only parasitic panel, 380 other indications than diarrhoea. ^Diarrhoea, nausea, vomiting or abdominal pain.

addressed to their parents. Among those who were invited, 462 consented to participate in the study, and we received a faecal sample from 438 of those. Using reported gastrointestinal symptoms in the week prior to faecal sample collection as the only exclusion criterion, we included 375 healthy controls in the study.

We registered sex, age, urban versus non-urban residency, and travel history to a foreign country from the request forms (patients with diarrhoea) and questionnaires (healthy controls). If residency was not stated on the request forms, we obtained this information from the laboratory records. We registered "No travel mentioned" if the request forms lacked this information. For patients with diarrhoea, the presence of EAEC and other intestinal pathogens and the cycle threshold (CT) values of the EAEC PCR were obtained from the laboratory records.

We used data from the national statistical institute of Norway, Statistics Norway (SSB), to compare the demographics of the study population to the general population in the county.

## Travel destinations

We grouped travel destinations in regions according to the Statistics Division of the United Nations Secretariat [50].

## Laboratory methods

**Sample collection.**   Faecal samples from patients with diarrhoea were collected in Copan FecalSwab® (Copan Diagnostics, Murrieta, USA) and/or sterile containers and received at the laboratory at the same day as collection from hospitalised patients and within one to three days from outpatients. Healthy controls sent faecal samples collected in Copan FecalSwab® transport medium via mail with transport time varying from one to three days.

**Pre-treatment.**   We mixed two-three drops or a pea-sized sample of faeces in a buffer made from 200 µL Molecular Grade Water (MGW) and 200 µL NucliSENS® Lysis Buffer (bioMérieux, Marcy-l'Étoile, France) and kept it at -20˚C overnight. The same volume of faeces was inoculated in 5 mL of Difco™ Selenite Broth and incubated at 35˚C for approximately

12 hours. The following day we defrosted, vortexed, and centrifuged the frozen sample, and mixed 200 μL of the supernatant with 50 μL from the incubated selenite broth into 2 mL of NucliSENS® Lysis Buffer. This mixture was vortexed and incubated in room temperature for at least 10 minutes.

## DNA extraction and PCR analysis

At the time of the study, the department routinely investigated faecal samples with the commercial real-time PCR gastro-intestinal panels Seegene Allplex$^{TM}$ GI-Bacteria(I) Assay, Seegene Allplex$^{TM}$ GI-Bacteria(II) Assay, Seegene Allplex$^{TM}$ GI-Virus Assay (Seegene Inc., Seoul, South Korea), and RIDA®GENE Parasitic Stool Panel II (R-Biopharm AG, Darmstadt, Germany). We performed one extraction for all panels. We added 10 μL bacterial Internal Control (IC), 10 μL viral IC, 10 μL MGW and 140μL of magnetic silica to the sample lysis buffer. This mixture was subjected to RNA and DNA extraction on a NucliSENS®easyMAG® (BioMerieux) instrument. Ridagene parasite IC was added after extraction. The PCR analyses were run on a Bio-Rad CFX96$^{TM}$ instrument (Bio-Rad, Hercules, USA). Both the Seegene and Ridagene panels show results in amplification curves and cycle threshold (CT) value and provide an automated interpretation as positive or negative. Both panels include positive controls for all pathogens. The *aggR* gene was the target gene for the EAEC PCR. We registered the presence of *Shigella*/EIEC, *Campylobacter jejuni/coli*, *Salmonella* spp, *Yersinia enterocolitica*, *Vibrio* spp, *Clostridioides difficile* toxin B, *Aeromonas* spp, EAEC, EHEC, EPEC, ETEC, Adenovirus genotypes (gt) 40 and 41, Astrovirus, Sapovirus, Rotavirus, Norovirus genotype 1 and 2, *Giardia lamblia*, *Cryptosporidium* spp and *Entamoeba histolytica*. We investigated faecal samples from healthy controls with the same PCR panels.

The department did not identify any *Vibrio cholerae* or *Entamoeba histolytica* in the study period and usually withheld positive *Aeromonas* PCR results from the report due to their ambiguous clinical significance. We excluded all three agents in our analyses.

## Statistical analysis

We performed all statistical analyses in R version 4.3.0., packages ggstatsplot_0.12.2, rstatix_0.7.2, and effectsize_0.8.6. We used Pearson chi-squared test to compare dichotomous variables, Kruskal-Wallis test to compare age in three groups or more, and Mann-Whitney U test to compare CT values or age between two groups. Whenever more than two groups were compared, we also performed posthoc tests with Bonferroni correction. We considered p-values <0.05 as significant.

## Ethics

The study was approved by the Regional Committee for Medical and Health Research Ethics (REK nord 2016/442).

## Results

### Characteristics of the study populations; pathogens detected, travel history, sex, age, residency, and seasonality

We examined 9487 samples from patients with diarrhoea and 375 samples from healthy controls. One or more pathogens were detected in 3832 (40.4%) of patients with diarrhoea and 57 (15.2%) of healthy controls (S1 Table). More than one pathogen was detected in 9.7% of samples from patients with diarrhoea, compared to 1.3% of healthy controls. The maximum number of pathogens detected in a single sample was five (n = 16). Travel history was mentioned in

**Table 1. Distribution of sex and age in patients with diarrhoea (diarrhoeal episodes), general population and healthy controls.**

| | Diarrhoeal episodes n = 9487 | | Healthy controls n = 375 | | General population n = 317363* | |
|---|---|---|---|---|---|---|
| Sex^ | | | | | | |
| Female | 5153 | (54.3%) | 222 | (59.2%) | 156207 | (49.2%) |
| Male | 4334 | (45.7%) | 153 | (40.8%) | 161156 | (50.8%) |
| Age (years)% | | | | | | |
| Median | 38 | | 55 | | 37 | |
| 0–9 | 1817 | (19.2%) | 39 | (10.4%) | 37619 | (11.9%) |
| 10–19 | 645 | (6.8%) | 24 | (6.4%) | 37075 | (11.7%) |
| 20–29 | 1441 | (15.2%) | 22 | (5.9%) | 50279 | (15.8%) |
| 30–39 | 997 | (10.5%) | 27 | (7.2%) | 42608 | (13.4%) |
| 40–49 | 914 | (9.6%) | 43 | (11.5%) | 43219 | (13.6%) |
| 50–59 | 902 | (9.5%) | 56 | (14.9%) | 38774 | (12.2%) |
| 60–69 | 1015 | (10.7%) | 81 | (21.6%) | 32936 | (10.4%) |
| 70–79 | 968 | (10.2%) | 72 | (19.2%) | 22211 | (7.0%) |
| 80–89 | 494 | (5.2%) | 9 | (2.4%) | 10147 | (3.2%) |
| >89 | 294 | (3.1%) | 2 | (0.5%) | 2495 | (0.8%) |
| Residency | n = 7993# | | | | | |
| Urban | 4779 | (59.8%) | 233 | (62.1%) | 190464 | (60%) |
| Non-urban | 3214 | (40.2%) | 142 | (37.9%) | 126899 | (40%) |

*Data from Statistics Norway (SSB) 2017. ^The difference in sex proportion is significant in pairwise analysis of diarrhoeal episodes and general population, and healthy controls and general population, with p-values <0.05. %The age difference is significant between diarrhoeal episodes and healthy controls (p<0.01), the general population was not included in this analysis. #1494 diarrhoeal episodes were not included because of permanent residency outside of Sør-Trøndelag County.

the request form for 1377 (14.5%) of the samples from patients with diarrhoea, and 18 (4.8%) healthy controls reported recent travel in the questionnaire (S2 Table).

There was a higher proportion of females among both patients with diarrhoea and healthy controls compared to the general population of Sør-Trøndelag County in 2017 (Table 1). The median age was 38 and 55 years in the cohorts of diarrhoeal episodes and healthy controls, respectively, compared to 37 years in the general population. In comparison with the general population, there was a proportionally larger number of persons in the youngest (0–9) and oldest ($\geq$ 80) age groups among those with diarrhoea, and the age groups 60–79 among healthy controls. The proportion of subjects residing in an urban setting was similar in both study cohorts and the general population.

There was no seasonal variability in the number of faecal samples from patients with diarrhoea received at the laboratory, with 23.6% of all samples received in winter (December–February), 25.4% in spring (March–May), 25.3% in summer (June–August), and 26.3% in autumn (September–November). For healthy controls, we obtained 27.5% of samples in winter, 17.9% in spring, 27.2% in summer, and 27.5% in autumn (S3 Table).

## Prevalence of EAEC

We detected EAEC in 440 (4.6%) of patients with diarrhoea and 8 (2.1%) healthy controls (Table 2). Among those with diarrhoea, EAEC was detected in all age groups. The age group prevalences ranged from 1.5% to 8.8%, with the highest prevalence in the age group 20–29 and the lowest in the age groups >79. In healthy controls, EAEC was only present in the age groups 40–49 and 60–79. The EAEC PCR CT values ranged from 14 to 43 in EAEC-positive samples from patients with diarrhoea and from 17 to 40 in EAEC-positive samples from healthy

**Table 2. The EAEC prevalence and median EAEC PCR CT values in patients with diarrhoea (diarrhoeal episodes) and healthy controls.**

|  | Diarrhoeal episodes | | Healthy controls | |
|---|---|---|---|---|
|  | Number tested | EAEC-positive | Number tested | EAEC-positive |
| **Total** | 9487 | 440 (4.6%) | 375 | 8 (2.1%) |
| **Sex** |  |  |  |  |
| Female | 5153 | 239 (4.6%) | 222 | 5 (2.3%) |
| Male | 4334 | 201 (4.6%) | 153 | 3 (2.0%) |
| **Age group** |  |  |  |  |
| 0–9 | 1856 | 58 (3.1%) | 39 | 0 |
| 10–19 | 669 | 29 (4.3%) | 24 | 0 |
| 20–29 | 1463 | 129 (8.8%) | 22 | 0 |
| 30–39 | 1024 | 56 (5.5%) | 27 | 0 |
| 40–49 | 957 | 43 (4.5%) | 43 | 1 (2.3%) |
| 50–59 | 958 | 48 (5.0%) | 56 | 0 |
| 60–69 | 1096 | 41 (3.7%) | 81 | 4 (4.9%) |
| 70–79 | 1040 | 24 (2.3%) | 72 | 3 (4.2%) |
| >79 | 799 | 12 (1.5%) | 11 | 0 |
| **Travel abroad** |  |  |  |  |
| Yes | 1371 | 262 (19.1%) | 18 | 0 |
| No | 8116 | 178 (2.2%) | 357 | 8 (2.2%) |
| **EAEC PCR CT value** |  |  |  |  |
| Median |  | 22 |  | 20.5 |
| Range |  | 14–43 |  | 17–40 |

controls, and their median CT values were 22 and 20.5, respectively. There was no difference in EAEC prevalence between males and females between patients with diarrhoea and healthy controls. The monthly prevalence in patients with diarrhoea ranged from 7.2% in August to 2.4% in June (S3 Table).

We detected EAEC in 262 (19.1%) of the 1371 samples from patients with diarrhoea and a reported travel history (S2 Table). The detection rate was highest in travellers to Africa (44.1%), Latin, Central, or South America (33.6%) and Asia (24.8%) (Fig 2). Among those with no history of travel, EAEC was equally prevalent in samples from participants with diarrhoeal episodes and healthy controls (2.2%). None of the 18 healthy controls who reported travel were EAEC-positive.

## EAEC in patients with diarrhoea

The majority (n = 440, 98.2%) of positive EAEC samples came from patients with diarrhoea. The proportion of females was 54.3% among patients with diarrhoeal episodes both with and without EAEC (S4 Table). The median age was 30 years in the group of EAEC-positive patients with diarrhoea, and 38 years in the group of EAEC-negative diarrhoeal episodes. Recent travel history was reported in 59.5% of those with diarrhoea and EAEC and in 12.3% of those with diarrhoea and no EAEC.

We detected one or more concomitant pathogens in 288 (65.5%) of EAEC-positive samples from patients with diarrhoea. Among these, 152 (52.8%) had one co-detection, 86 (29.9%) had two co-detections, 34 (11.8%) had three co-detections, and 16 (5.6%) had four co-detections. The most commonly detected concomitant pathogens were EPEC (n = 127, 44.1%) and ETEC (n = 109, 37.8%) (Fig 3). Of the 3441 EAEC-negative samples from patients with diarrhoea with at least one pathogen detected, 634 (18.4%) had more than one pathogen. The proportion

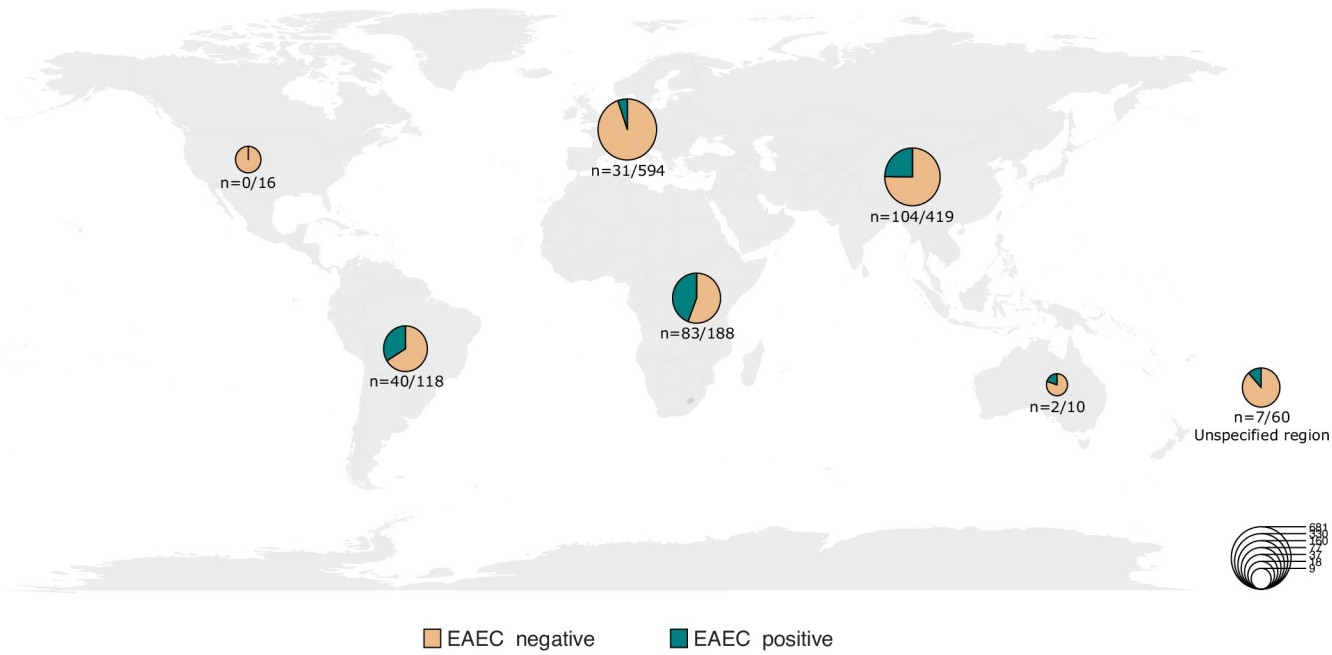

**Fig 2. EAEC positive and EAEC negative samples from patients with diarrhoea (diarrhoeal episodes) and travel history by region\*.** \*Pie chart size represents total number of travellers.

of concomitant pathogens rose to 79.3% when looking only at EAEC-positive episodes with recent travel history.

## EAEC-positive with concomitant pathogens, EAEC-positive alone, and EAEC-negative patients with diarrhoea

The proportion of persons with urban residency was higher in the group of EAEC-positive diarrhoeal episodes with concomitant pathogens compared to the EAEC-negative diarrhoeal

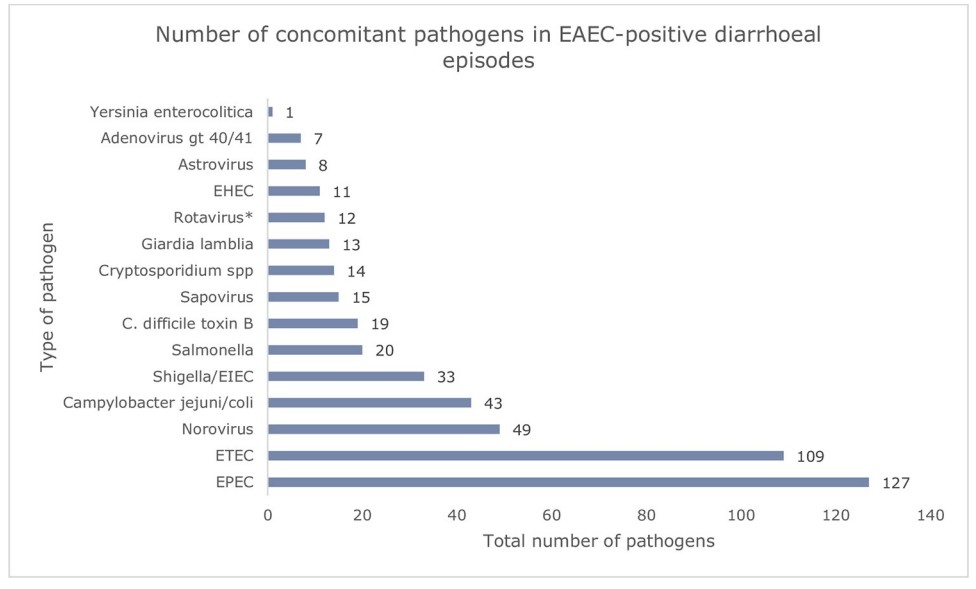

**Fig 3. Concomitant pathogens in EAEC positive samples from patients with diarrhoea.**

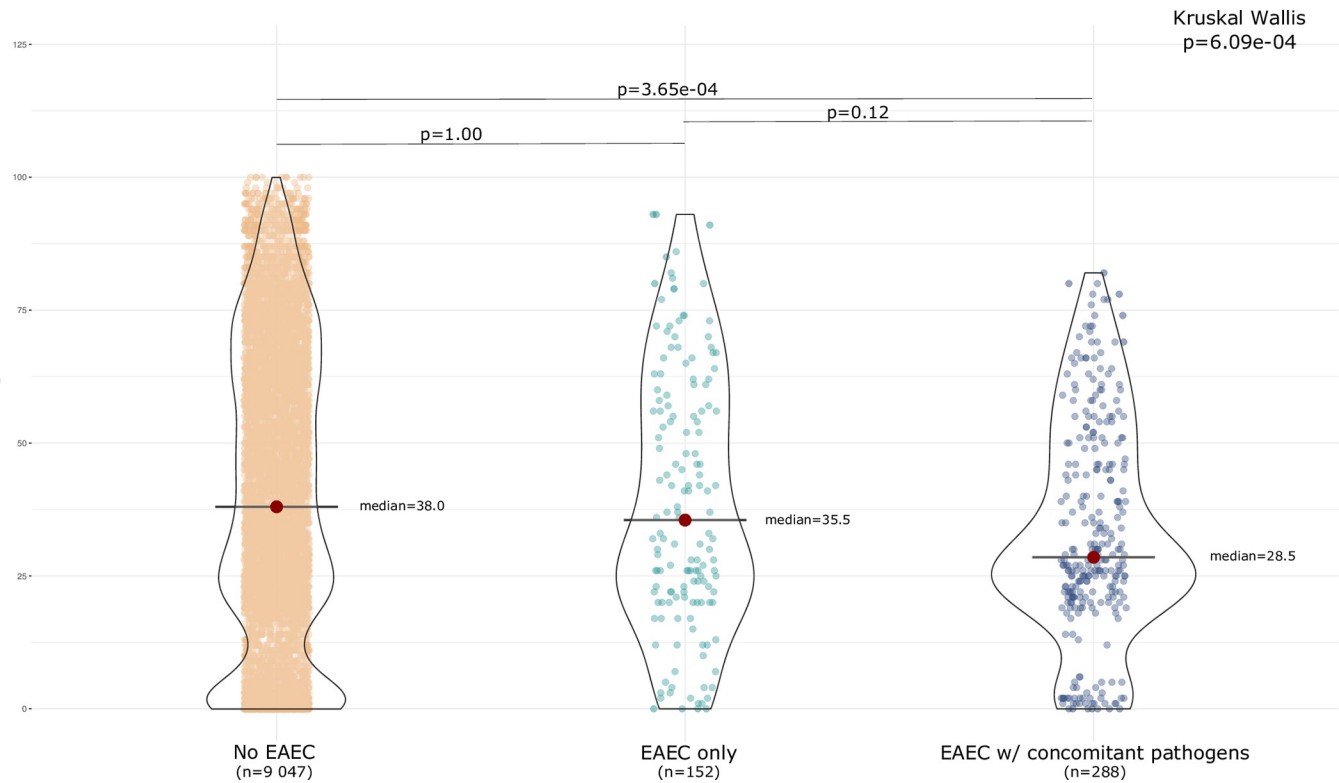

**Fig 4. Age in patients with diarrhoea with No EAEC vs EAEC only vs EAEC with concomitant pathogens.**

episodes (p < 0.01) (S5 Table). The group with EAEC and concomitant pathogens were younger than those with no EAEC (p < 0.01) (Fig 4). Though there was a trend towards younger age in those with EAEC with concomitant pathogens compared to those with EAEC only, and in those with EAEC only compared to those with no EAEC, these differences were not significant.

Recent travel was more commonly reported in those with EAEC and concomitant pathogens compared to the two other groups (p < 0.01), and in those with only EAEC compared to those with no EAEC (p < 0.01) (Fig 5).

We found no difference in the sex distribution when comparing EAEC-positive diarrhoeal episodes with and without concomitant pathogens and EAEC-negative diarrhoeal episodes (S5 Table). The distribution and median of EAEC PCR CT values were similar in those with and without concomitant pathogens.

## Discussion

### EAEC prevalence

In our study, EAEC was a commonly detected pathogen in both patients with diarrhoea and healthy controls. The prevalence was more than twice as high in those with diarrhoea compared to the control group. We found no difference in EAEC prevalence between males and females. The proportion of EAEC-positive persons differed between age groups, and the highest prevalence was found in young adults with diarrhoea. It is possible that persons in this age group have travel habits and behaviours that put them at increased risk of being infected with EAEC. In healthy controls, EAEC was only found in the age groups 40–49 and 60–79. Due to

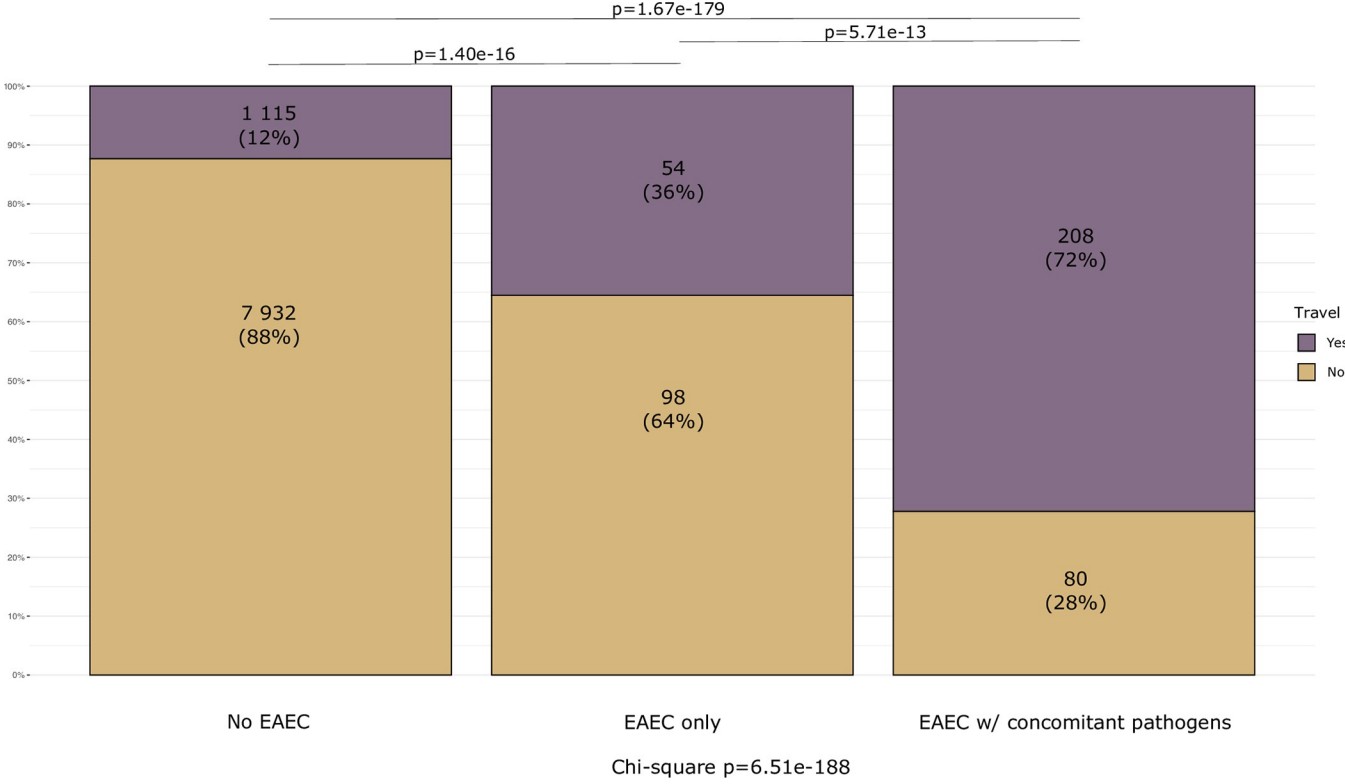

**Fig 5. Recent travel in No EAEC vs EAEC only vs EAEC with concomitant pathogens.**

the low number of EAEC-positive samples, as well as in some age groups, we cannot make conclusions from the age distribution in healthy controls.

The prevalence of EAEC among travellers was significantly higher than in non-travellers. The observed difference in EAEC prevalence between patients with diarrhoea and healthy controls disappeared when travel history was excluded from the analysis, which may suggest that EAEC is not associated with diarrhoea in the absence of recent travel to a foreign country.

The EAEC prevalence among healthy controls in our study is similar to other studies from industrialised regions, but comparisons are difficult due to differences in EAEC detection methods and heterogeneity of population characteristics and study designs. The detection rate among healthy controls in our study is in concordance with the British IID study where a re-examination of frozen stool samples detected EAEC in 2% of 2205 matched controls using real time PCR targeting the *aat* gene [51]. They found a prevalence of 6% among diarrhoeal cases, which was slightly higher than ours. An American publication reported 3159 (3.5%) EAEC positive samples among 91401 clinical samples from 45 different laboratories analysed with a multiplex PCR method targeting EAEC genes *aggR* and *aat* [52]. Clinical data and information on whether multiple samples could be from the same person was lacking.

Compared to our study, a Finnish study using PCR targeting *aggR* alone and a Danish study using PCR targeting *aggR*, *aatA*, and *aaiC* both found slightly lower EAEC prevalences in study participants without diarrhoea of 1% and 1.2%, respectively [39, 53, 54]. The healthy volunteers of the Finnish study were recruited from a travel clinic prior to their travel, and the Danish study found volunteers among university employees, students and subjects, submitting a stool sample in conjunction with a general health examination.

The above-mentioned Finnish study and a German study on travellers' diarrhoea, both report an EAEC prevalence of 45% in returned travellers with diarrhoea [38]. This is considerably higher than the prevalence among travellers with diarrhoea in our study, but their study designs might have selected more high-risk travellers as they included cases from a travel clinic in Helsinki and a Department of Infectious Diseases and Tropical Medicine in Munich, respectively. We detected a similar prevalence among the travellers with diarrhoea who had visited Africa, with 83 of 188 (44.1%) being EAEC-positive.

A variety of genetic targets have been used for the detection of EAEC, largely because a molecular definition is lacking. A publication by Boisen et al. characterising 97 EAEC strains using HEp-2 cell assay and whole genome sequencing found the *aggR* gene in 84% of isolates, making it the second most common AggR regulon-associated gene after the dispersin gene *aap*, which was found in 89% [10]. Two other gene targets used for the identification of EAEC, *aatA* and *aaiC*, were present in 75% and 38%, respectively.

A volunteer study where only one of four EAEC strains resulted in diarrhoea [15] and outbreaks caused by specific EAEC strains [44–46] support the presumption that different EAEC strains have different pathogenic potential, but there is still insufficient evidence to conclude on the best genetic marker to identify the more pathogenic strains. Boisen et al. detected all of the above-mentioned genes more commonly in EAEC isolates from healthy controls than from diarrhoeal cases. Any study's choice of gene targets for detection can impact not only the recorded EAEC prevalence but also the association between EAEC and diarrhoea.

## EAEC and concomitant pathogens

We detected concomitant pathogens in the majority (65.5%) of EAEC-positive samples from patients with diarrhoea. Our findings are similar to those in an American study, which detected concomitant pathogens in more than 60% of the EAEC-positive samples [52].

There is little knowledge on the interplay between potentially enteropathogenic microbes in cases of infectious diarrhoea. The importance of concomitant pathogens in EAEC infection is also poorly understood, and studies on EAEC in diarrhoeal disease often include investigations of a limited number of other pathogens [22, 36, 55].

Though there are some studies reporting results from analyses of broad panel molecular methods used on faecal samples have in recent years, there are few reports on the clinical implication of concomitant pathogens in EAEC infection. Comparisons with such studies are often difficult either because they do not report the presence of concomitant pathogens or do not include EAEC in their analyses, possibly due to the ambiguous clinical impact of EAEC [56–58]. One North American and one recent Danish publication present results from syndromic testing with multiplex PCR assays performed on large numbers of faecal samples and include coinfections in their results. They both report coinfections in more than 60% of EAEC-positive samples, which is similar to our study [49, 52]. Neither study describes the characteristics of EAEC-positive samples.

In our study, the patients with diarrhoea with EAEC-positive samples and concomitant pathogens were younger and had a higher proportion of urban residency compared to the EAEC-negative patients with diarrhoea. They also more often reported recent foreign travel compared to both the EAEC-negative group and those with EAEC without concomitant pathogens. Adachi *et al.* found concomitant pathogens in less than half of travellers with EAEC-positive diarrhoea, which is a lower proportion than in our study, but they used immunoassay and culture for detection of bacteria and did not identify viruses [59]. In the studies by Lääveri and Paschke *et al.*, two or more pathogens were detected in 53% and 60.5% of travellers with

ongoing diarrhoea respectively, but none specified the prevalence of concomitant pathogens in EAEC-positive samples [38, 39].

The median EAEC PCR CT value was similar in samples with concomitant pathogens as in samples without. The CT value was also similar between EAEC-positive diarrhoeal episodes and EAEC-positive healthy controls, indicating that PCR-based semi-quantitative evaluation of bacterial concentration by use of the CT value cannot separate symptomatic from asymptomatic EAEC infection. This finding contrasts with the significant association between CT value and disease state reported by Chattaway *et al.* [60]. Few studies have examined the clinical significance of EAEC PCR CT values.

The high proportion of concomitant pathogens can imply that EAEC is an innocent bystander or may simply reflect that the pathogens share a common risk factor such as recent travel to low-income countries. It is also possible that intestinal pathogens act in synergy by facilitating adhesion, invasion, or proliferation of other pathogens, or by potentiating each other's toxic or inflammatory effects. Izquierdo *et al.* suggests that the gut microbiota affects the pathogenicity of diarrhoeagenic *E. coli* and found that supernatants from *Citrobacter werkmanii* and *Escherichia albertii* upregulated several virulence factors in STEC and EAEC in vitro [61]. Similar interactions may occur between diarrhoeal pathogens. The possibility that EAEC is part of a polymicrobial gastrointestinal infection should be considered and further investigated.

## Strengths and limitations

The large number of samples from patients with diarrhoea included and the longitudinal time aspect to cover seasonal variations are among the strengths of this study. Using randomly invited controls might reduce selection bias and examining each faecal sample for several major gastrointestinal pathogens makes possible a more comprehensive evaluation.

Among the limitations of this study are the low response rate from invited volunteers for the control group, and the selection bias that occurs in the process of volunteering to participate. The median age was higher in healthy controls compared to both patients with diarrhoea and the general population in the region and had a higher proportion of female sex compared to the general population.

The request forms for faecal samples from patients with diarrhoea sometimes lacked relevant information and some of the samples could have been collected for other reasons than diarrhoea. We expect this to have happened in a minority of cases, as diarrhoea is the only accepted indication for performing all three multiplex PCR panels. Due to the lack of clinical information some of the diarrhoeal episodes might have been wrongly categorised with no recent foreign travel.

EAEC was present in only eight healthy controls, which limits the value of our comparison of EAEC-positive symptomatic and asymptomatic individuals. We have no follow-up information on EAEC-positive healthy controls, and there is a possibility that some of these developed diarrhoea after participating in the study.

## Conclusions

In this study, EAEC is a common detection in faecal samples from patients with diarrhoea who have undertaken recent foreign travel. Our findings indicate that domestically acquired EAEC is not associated with diarrhoea. Most EAEC-positive patients with diarrhoea are co-infected with other intestinal pathogens, which may suggest that clinical disease result from an interaction between several pathogens. Those with EAEC-positive diarrhoea and concomitant pathogens are more likely to be young, have urban residency and a recent travel history to a

foreign country. The EAEC PCR CT value does not seem to be useful in determining an association between detecting EAEC and having diarrhoea.

Further studies into bacterial and host genetics in order to evaluate the role of EAEC in diarrhoeal disease are called upon, and one should also consider interactions between pathogens in these studies.

## Supporting information

**S1 Table. Pathogens detected in patients with diarrhoea (diarrhoeal episodes) and healthy controls.**
(DOCX)

**S2 Table. Travel destinations in patients with diarrhoea (diarrhoeal episodes) and healthy controls and EAEC-positive diarrhoeal episodes and healthy controls.** *Travel to more than one continent included in numbers; 22 of the diarrhoeal episodes and 1 of the healthy controls travelled to more than one continent ^Regions as defined by the Statistics Division of The United Nations Secretariat # Prevalence of EAEC in travellers to same region.
(DOCX)

**S3 Table. Total number of, and EAEC-positive, samples received from patients with diarrhoea (diarrhoeal episodes) and healthy controls by year and month.** *Diarrhoeal episodes, ¤EAEC-positive Diarrhoeal episodes, #HC = Healthy controls, ‰EAEC-positive Healthy controls, &proportion of EAEC-positive Diarrhoeal episodes, +proportion of EAEC-negative Healthy controls.
(DOCX)

**S4 Table. Characteristics of patients with diarrhoea with and without EAEC.**
(DOCX)

**S5 Table. Characteristics of patients with diarrhoea with EAEC and concomitant pathogens, EAEC and no concomitant pathogens, and no EAEC.** *We only had information on urban or non-urban residency in 233 EAEC-positive with concomitant pathogens, 121 EAEC positive without concomitant pathogens, and 7639 EAEC-negative diarrhoeal episodes ^Only EAEC-positive with concomitant pathogens versus EAEC-negative were significant with p<0.05 in pairwise analyses.
(DOCX)

## Acknowledgments

The authors would like to thank the laboratory technicians at the Department of medical microbiology, St. Olavs Hospital, for helping with sample logistics and analyses.

## Author Contributions

**Conceptualization:** Marit Gudrun Husby, Heidi Lange, Jan Egil Afset.

**Data curation:** Ingvild Haugan.

**Formal analysis:** Ingvild Haugan, Lene Christin Olsen, Melanie Rae Simpson, Heidi Lange, Jan Egil Afset.

**Funding acquisition:** Ingvild Haugan, Jan Egil Afset.

**Investigation:** Ingvild Haugan, Marit Gudrun Husby, Bjørg Skjøtskift, Dorothea Aamnes Mostue, Andreas Brun.

**Methodology:** Ingvild Haugan, Marit Gudrun Husby, Bjørg Skjøtskift, Dorothea Aamnes Mostue, Andreas Brun, Melanie Rae Simpson, Heidi Lange, Jan Egil Afset.

**Project administration:** Ingvild Haugan, Jan Egil Afset.

**Supervision:** Jan Egil Afset.

**Visualization:** Lene Christin Olsen.

**Writing – original draft:** Ingvild Haugan.

**Writing – review & editing:** Ingvild Haugan, Marit Gudrun Husby, Dorothea Aamnes Mostue, Lene Christin Olsen, Melanie Rae Simpson, Heidi Lange, Jan Egil Afset.

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
