## [Decision Letter · Decision Letter 0]

9 Jan 2024

PONE-D-23-26927Enteroaggregative Escherichia coli in persons with diarrhoea and healthy controls in mid-Norway: A prospective, case control studyPLOS ONE

Dear Dr. Haugan,

Thank you for submitting your manuscript to PLOS ONE. After careful consideration, we feel that it has merit but does not fully meet PLOS ONE’s publication criteria as it currently stands. Therefore, we invite you to submit a revised version of the manuscript that addresses the points raised during the review process.

We look forward to receiving your revised manuscript.

Kind regards,

Furqan Kabir

Academic Editor

PLOS ONE

Journal Requirements:

Did you know that depositing data in a repository is associated with up to a 25% citation advantage (https://doi.org/10.1371/journal.pone.0230416)? If you’ve not already done so, consider depositing your raw data in a repository to ensure your work is read, appreciated and cited by the largest possible audience. You’ll also earn an Accessible Data icon on your published paper if you deposit your data in any participating repository (https://plos.org/open-science/open-data/#accessible-data).

"This research was financially supported by the Clinic of Laboratory Medicine, St. Olavs Hospital (grant numbers 17/10862-27 and 18/11252-11). The funders had no role in study design, data collection and analysis, decision to publish, or preparation of the manuscript."

We note that one or more of the authors is affiliated with the funding organization, indicating the funder may have had some role in the design, data collection, analysis or preparation of your manuscript for publication; in other words, the funder played an indirect role through the participation of the co-authors. If the funding organization did not play a role in the study design, data collection and analysis, decision to publish, or preparation of the manuscript and only provided financial support in the form of authors' salaries and/or research materials, please do the following:

(1) Review your statements relating to the author contributions, and ensure you have specifically and accurately indicated the role(s) that these authors had in your study. These amendments should be made in the online form.

(2) Confirm in your cover letter that you agree with the following statement, and we will change the online submission form on your behalf: 

**Additional Editor Comments:**

Please go through the reviewers' comments carefully and draft your responses accordingly. 

Reviewers' comments:

Reviewer's Responses to Questions

**Comments to the Author**

1. Is the manuscript technically sound, and do the data support the conclusions?

Reviewer #1: Partly

Reviewer #2: Partly

Reviewer #3: Yes

Reviewer #4: Yes

Reviewer #5: Yes

Reviewer #6: Partly

2. Has the statistical analysis been performed appropriately and rigorously? 

Reviewer #1: No

Reviewer #2: N/A

Reviewer #3: Yes

Reviewer #4: Yes

Reviewer #5: Yes

Reviewer #6: I Don't Know

3. Have the authors made all data underlying the findings in their manuscript fully available?

Reviewer #1: Yes

Reviewer #2: Yes

Reviewer #3: Yes

Reviewer #4: Yes

Reviewer #5: Yes

Reviewer #6: Yes

4. Is the manuscript presented in an intelligible fashion and written in standard English?

Reviewer #1: Yes

Reviewer #2: Yes

Reviewer #3: Yes

Reviewer #4: Yes

Reviewer #5: Yes

Reviewer #6: No

5. Review Comments to the Author

Reviewer #1: Review PONE-D-23-26927

TITLE: Enteroaggregative Escherichia coli in persons with diarrhoea and healthy controls in mid-Norway: A prospective, case control study

The study describes the EAEC findings from patients at a university hospital in Mid-Norway with comparisons to healthy subjects.

The findings in the control group are, as the authors themselves comment, less frequently described in a group with no immediate intention to travel; in traveller studies the pre-travel samples have mostly been collected at travel clinics, and those with an intention to travel are also likely to have travelled in the past. However, Bruijnesteijn van Coppenraet et al. have a rather similar study setting in the Netherlands in their 2015 article.

The main problem with the study lies in the control group setting. Nontravellers should not be used as controls: Among asymptomatic travellers, EAEC is found in 16-28% of stool samples (Paschke et al. 2011, Lääveri et al. 2016, Lertsethtakarn-Ketwalha et al. 2018). By contrast, as the authors observe, EAEC is a rare finding in pre-travel stool samples in many traveller studies.

The authors state that nontravellers were defined as not having travel history documented. This does not automatically equal to having no travel history since unless this was actively documented as “no travel history” instead of information on travel history missing i.e. not documented at all.

To my knowledge, the interplay between the various gastrointestinal pathogens has been poorly described in clinical settings particularly among travellers. Furthermore, the association of EAEC with age is not independent from history of travel to low- and middle-income countries; EAEC can be found in stools for several weeks after travel. Therefore, in addition to using nontravellers as controls, I don’t agree with the use of a multivariable regression model used for the identification of the role of EAEC. A more practical approach would have been “only EAEC” vs. “no EAEC” vs “EAEC together with other pathogens. In addition, asymptomatic colonization with Salmonella and even ETEC is relatively common among travellers; this has been described in many studies with control group design (e.g. Paschke et al.2011, Lääveri et al.2016, Lertsethtakarn-Ketwalha et al. 2018). Moreover, the aetiology of diarrhoea among nontravellers is generally considered to be of viral origin although norovirus findings among travellers are more frequent in more recent traveller studies than e.g. 30 years ago.

Unfortunately, due to the issues mentioned above, I cannot recommend accepting this article. However, I encourage the authors to change this into a more descriptive study instead of trying to assess the role of EAEC in diarrhoea; for that purpose they would need to analyze travellers and nontravellers separately and the proportion of travellers in the control group appears too small for statistical analyses.

Reviewer #2: 1. The title needs to be modified, as the case control study itself reflects the inclusion of healthy as well as infected individuals, the usage twice may be avoided.

2. The geo co-ordinates of the study setting may be given for appropriate location.

3. Cite suitable references for methodology. For instance, the sample processing, isolation of EAEC and molecular detection. Alike, the gene sequence and PCR conditions.

4. ‘maximum number of pathogens detected in a single sample was five (n=16)’: interesting to find it in text.

5. Representation of the data year wise or season wise would also fetch more interesting findings.

6. Minor typos and grammatical errors need to be addressed.

Reviewer #3: The study was well designed, and the manuscript is generally well-written and I nearly found no concern. At the discretion of the editor, authors should clarify on the following;

- There is no justification / rationale for conducting the study in mid-Norway, in Sør-Trøndelag County

- You did one extraction for the assays targeting both nucleic acids? Yet, some viruses targeted e.g. Astrovirus, Sapovirus etc are RNA so clarify on whether the kit achieves extraction of RNA and DNA simultaneously.

Reviewer #4: Please note : There is no novelty in the study. The fact of EAEC associated with travellers diarrhoea is known . The study undertaken on huge sample size highlights the prevalence in that region. What is the additional information gained .The clinical presenation and duration of illness is not mentioned. The treatment and follow up of the patients is not looked into.There are kits for multiplex PCR on the stool to pick up the pathogens . There is no new finding other than the prevalence of EAEC in that region.

Reviewer #5: Well done on the well written piece of work. Just one or two comments

1) On Lines 267 – 270 unnecessary speculation. If it was not done why mention it?

2) Figure 3 - what is the unit on the Y-axis? What exactly is this Figure showing? And can the results be explained for the 60 -69 & 70 - 79 age groups.

3) All Figures seem to be missing legends.

Reviewer #6: I have reviewed the manuscript and found several grammar mistakes that should be corrected. Below are some specific points that require attention

Methods: Study populations

1. The age group of patients and healthy individuals included in the study should be described.

2. Line 105-106: “multiple samples from the same person as duplicates” The authors need to justify why they excluded repeat samples from the same patient, given that gastro pathogens are known to shed intermittently in the stool. A repeat sample is generally recommended in diarrheal cases to increase the chances of detecting a pathogen. Also, what is the rationale for selecting a 3-month window period?

3. Line 107: The meaning of "did not reveal a previously undiagnosed pathogen" is not very clear.

4. Line 117: exclusion criteria: Did the authors consider antibiotic consumption history in the control group before sample collection (as it can take up to four weeks for the normal flora to establish after a 7 days antibiotic course)? if so what period before sample collection was considered?

5. Did the authors follow up with the control group participants especially those who tested positive for a pathogen for the development of any diarrheal symptoms after the sample collection? Some of the control group participants may have been in the infection development state at the time of sample collection.

6. The authors need to briefly describe the real-time PCR parameters used, such as whether it was SYBR green-based/Taqman probe-based, and the machine used.

7. Did the authors use any positive controls or perform the assay in duplicates/triplicates to confirm the reproducibility of results?

Travel destinations:

8. The authors should specify the duration of travel that was considered a travel case. Also, was travel to a foreign country the only consideration, or was travel within the country also considered?

Sample collection:

9. Did the authors differentiate hospital-acquired diarrhoea from community-acquired diarrhoea? The authors should explain what the hospitalized patients represent.

10. The lysis buffer composition or brand name of the reagents used in the study where available should be provided.

Results:

11. Line 215: The authors should explain how they decided on the CT value cutoffs. Did the Multiplex PCR assay provide them?

12. The values described in Line 222-224 should be either ascending or descending.

13. The authors have not described whether any seasonal patterns were observed in the study but have included this as a strength of the study.

14. It would be interesting to know the authors' thoughts on the discovery of mosaicism of virulence factors in diarrheagenic E. coli pathotypes based on genomic studies.

Discussion:

15. The authors should consider reducing the number of discussion points.

6. PLOS authors have the option to publish the peer review history of their article (what does this mean?). If published, this will include your full peer review and any attached files.

Reviewer #1: No

Reviewer #2: No

Reviewer #3: **Yes: **David Patrick Kateete

Reviewer #4: No

Reviewer #5: No

Reviewer #6: No

---

## [Author Response · Author response to Decision Letter 0]

22 Feb 2024

Dear Reviewers,

Thank you for your comments, which allows us to improve our paper «Enteroaggregative Escherichia coli in persons with diarrhoea and healthy controls in mid-Norway: A prospective, case control study» now renamed to «Enteroaggregative Escherichia coli in mid-Norway: A prospective, case control study». We sincerely appreciate your effort, and we have attempted to revise our paper according to your suggestions for improvement. Please do not hesitate to contact us if any of our responses are unclear or if you want us to make additional changes.

Sincerely, 

first author of the manuscript on behalf of all co-authors

Reviewer 1

The main problem with the study lies in the control group setting. Nontravellers should not be used as controls: Among asymptomatic travellers, EAEC is found in 16-28% of stool samples (Paschke et al. 2011, Lääveri et al. 2016, Lertsethtakarn-Ketwalha et al. 2018). By contrast, as the authors observe, EAEC is a rare finding in pre-travel stool samples in many traveller studies. 

Author´s reply: Thank you for your time and effort in order to help us improving our paper. We are very grateful that you have taken time to present such concrete and practical suggestions for improvement. You have suggested a different approach to the statistical analyses, and to change the paper into a more descriptive study. We agree with the issues you point out and have tried to change the manuscript according to your suggestions. Since this is such a major change, causing changes in both the introduction, results, and discussion sections, we cannot name line numbers for these changes. 

We agree that there are problems in the control group setting. In addition to having different demographics to the general population and case cohort in regard to sex and age distribution, the number of healthy controls is very small when dividing into subgroups such as travellers and EAEC-positive samples. These are limitations of the study that we have tried to be transparent about, and we have attempted to address these issues in our discussion. We also agree that nontravellers should not be used as controls to travellers. As a result of the design and purpose of our study, we do not have access to pre- and post-travel samples from travellers. In our revised version of the manuscript, we have not performed any analyses comparing non-travellers to travellers.

The authors state that nontravellers were defined as not having travel history documented. This does not automatically equal to having no travel history since unless this was actively documented as “no travel history” instead of information on travel history missing i.e. not documented at all. 

Author´s reply: It is true that not having travel history specified on the request form excludes any recent foreign travel. We believe that it is quite likely that some non-travellers in the case cohort in fact have had recent foreign travel, and this might cause a falsely high prevalence of pathogens in the non-traveller group and underestimate the differences between these groups in our comparisons. This is a weakness in our study caused by limited access to clinical information. We cannot gain access to more clinical information but have addressed this in the discussion around the limitations of this study. Lines 393-393.

Therefore, in addition to using nontravellers as controls, I don’t agree with the use of a multivariable regression model used for the identification of the role of EAEC. A more practical approach would have been “only EAEC” vs. “no EAEC” vs “EAEC together with other pathogens. 

Author´s reply: We have removed the multivariable regression analysis and instead we describe the differences between the three groups mentioned and compare them in univariate analyses using Kruskal-Wallis test and Mann-Whitney U test.

However, I encourage the authors to change this into a more descriptive study instead of trying to assess the role of EAEC in diarrhoea; for that purpose they would need to analyze travellers and nontravellers separately and the proportion of travellers in the control group appears too small for statistical analyses.

Author´s reply: We agree that the number of travellers in the control group is too small for statistical analyses. We also agree that our study could perform better as a descriptive study and have attempted to rewrite the manuscript accordingly.

Reviewer 2

Thank you for your time and effort to help us improve our paper.

The title needs to be modified, as the case control study itself reflects the inclusion of healthy as well as infected individuals, the usage twice may be avoided.

Author´s reply: We fully agree and have made changes to the title as suggested. 

The geo co-ordinates of the study setting may be given for appropriate location.

Author´s reply: We are happy to accommodate that if editor finds it important. We are not sure how to write co-ordinates for such a large geographical area as mid-Norway and would be grateful for any advice on that. Alternatively, we could give the co-ordinates for our laboratory but are not familiar with that request for similar studies.

Cite suitable references for methodology. For instance, the sample processing, isolation of EAEC and molecular detection. Alike, the gene sequence and PCR conditions.

Author´s reply: We agree that references for methodology should be cited whenever possible. However, the procedure for sample pretreatment used in this study is an internally non-published validated method at the laboratory, while DNA extraction and PCR analysis were done according to protocol from the manufacturers, and which are available from the manufacturer. We therefore believe it should not be necessary to include detailed description of standard protocols in the manuscript. However, if the editor wants us to include such information we can do so. The classification of EAEC as present or absent in a sample in this study was based on PCR on DNA eluate from stools samples, not on isolation of EAEC in pure culture. We do not have access to detailed information about gene sequences target by this method. 

‘maximum number of pathogens detected in a single sample was five (n=16)’: interesting to find it in text.

Author´s reply: We agree with that this is an interesting detection. We interpreted this as a comment, not a suggestion to make changes in the manuscript.

Representation of the data year wise or season wise would also fetch more interesting findings.

Author´s reply: We agree that season wise representation can add useful and interesting findings and have included this in the manuscript. Lines 222-226 and 235-236.

Minor typos and grammatical errors need to be addressed.

Author´s reply: We have proofread the manuscript and corrected the errors we could find. 

Reviewer 3

Thank you for your time and effort to help us improve our paper.

There is no justification / rationale for conducting the study in mid-Norway, in Sør-Trøndelag County

Author´s reply: That is true. The geographical area is chosen simply because the laboratory the first and last authors work at analyse samples from persons living in this area, and these are the data we have access to. Our laboratory implemented broad panel PCR testing of faecal samples early compared to most other laboratories in Norway, which gave us access to data many other researchers would not be able to obtain. Even today, such broad panel testing is not widely used in laboratories. We have tried to emphasise this in the introduction of the new version. Lines 95-99.

You did one extraction for the assays targeting both nucleic acids? Yet, some viruses targeted e.g. Astrovirus, Sapovirus etc are RNA so clarify on whether the kit achieves extraction of RNA and DNA simultaneously.

Author´s reply: Thank you for pointing out that. We have clarified it in the revised manuscript. Lines 169-170.

Reviewer 4

Thank you for your time and effort to help us improve our paper.

There is no novelty in the study. The fact of EAEC associated with travellers diarrhoea is known . The study undertaken on huge sample size highlights the prevalence in that region. What is the additional information gained. (….) There is no new finding other than the prevalence of EAEC in that region.

Author´s reply: As far as we now, until July 2023 there was no published studies with results from such broad panel PCR testing of faecal samples in the Nordic countries. As of today we are aware of one such study as referred to in our manuscript (lines 93-95). We are also not aware of any other epidemiological study on EAEC that include adults from any Nordic country. We believe that these are useful results from a microbiological and diagnostic point of view. 

The clinical presentation and duration of illness is not mentioned. The treatment and follow up of the patients is not looked into.

Author´s reply: We agree that more clinical information, including clinical presentation, duration of illness, treatment and follow up, would be interesting to look at. Our study design includes a large dataset at the expense of detailed information. It has not been the aim of this study to look into details of clinical presentations and outcomes of diarrhoea in EAEC-positive persons. 

Reviewer 5

Thank you for your time and effort to help us improve our paper.

On Lines 267 – 270 unnecessary speculation. If it was not done why mention it?

Author´s reply: Thank you for addressing this unclarity in the manuscript. We wanted to discuss the issues of using different methodology for the identification of EAEC. The entire discussion section has now undergone a major revision, and the sentence that was found on lines 267-270 has been removed.

Figure 3 - what is the unit on the Y-axis? What exactly is this Figure showing? And can the results be explained for the 60 -69 & 70 - 79 age groups.

Author´s reply: We apologise for missing labels in the figures. This is corrected in the revised manuscript. As part of the revision of the manuscript, figure 3 is no longer included. We do not believe we can draw conclusions from the EAEC prevalence in age groups among healthy controls due to the low number. This is discussed in lines 294-296.

All Figures seem to be missing legends.

Author´s reply: We apologise for this mistake. This is corrected in the revised manuscript. 

Reviewer 6

Thank you for your time and effort to help us improve our paper.

I have reviewed the manuscript and found several grammar mistakes that should be corrected. 

Author´s reply: We have proofread the manuscript and corrected the errors we could find. 

The age group of patients and healthy individuals included in the study should be described.

Author´s reply: We agree. We have described the highlights of differences in ages in the text (lines 209-213) and the details on age groups are included in Table 1. 

Line 105-106: “multiple samples from the same person as duplicates” The authors need to justify why they excluded repeat samples from the same patient, given that gastro pathogens are known to shed intermittently in the stool. A repeat sample is generally recommended in diarrheal cases to increase the chances of detecting a pathogen. Also, what is the rationale for selecting a 3-month window period?

Line 107: The meaning of "did not reveal a previously undiagnosed pathogen" is not very clear.

Author´s reply: We have tried to rephrase so it is clearer that any samples with new detections are included as separate episodes, while repeated samples within 3 months with no new pathogens detected are considered same episode and excluded (Lines 118-122). Since the DNA and RNA of pathogens can be shed in stool for a long time after an episode of infectious diarrhoea, we chose a window period of 3 months. There is a lack of consensus on what time period would be the best to define when a detection represents a new infection or remnants from a previous infection.

exclusion criteria: Did the authors consider antibiotic consumption history in the control group before sample collection (as it can take up to four weeks for the normal flora to establish after a 7 days antibiotic course)? if so what period before sample collection was considered?

Author´s reply: We have not taken into consideration any antibiotic consumption. We agree that information on the use of antibiotics could give interesting information since it will have an effect on the gastrointestinal microbiota. We lack knowledge on how antibiotics will affect EAEC in the intestines and have chosen to look at the presence of EAEC in controls without diarrhoea independent of recent consumption of antibiotics. At any time, a proportion of the general population will take antibiotics, and one can argue that one should not adjust for this in an epidemiological study. 

Did the authors follow up with the control group participants especially those who tested positive for a pathogen for the development of any diarrheal symptoms after the sample collection? Some of the control group participants may have been in the infection development state at the time of sample collection.

Author´s reply: We did not follow up the control group participants as this was not part of the original study design or among the aims of the study. However, we recognise that there is a possibility that EAEC-positive healthy controls were in an incubation period and would develop diarrhoea, and we agree that such a follow-up could have been of interest. The EAEC prevalence we find among healthy controls does not differ a lot from other epidemiological studies. However, this is a limitation of our study, and we have addressed that in our discussion on strengths and limitations. Lines 395-397.

The authors need to briefly describe the real-time PCR parameters used, such as whether it was SYBR green-based/Taqman probe-based, and the machine used.

Author´s reply: We have added information on the PCR instrument used (Line 171). The PCR methods are commercial methods. We do not have access to information on all PCR conditions. We have performed the analysis according to the manufacturers’ instructions and information can be found on their web sites. We have named the methods and manufacturers in the manuscript which is commonly done in studies that have used such commercial methods, and do not believe that it is necessary to provide further references on these methods.

Did the authors use any positive controls or perform the assay in duplicates/triplicates to confirm the reproducibility of results?

Author´s reply: We have performed the analyses according to the suppliers’ instructions, including positive controls for all pathogens. We have added this information in lines 173-174. We did not perform PCR analyses in duplicates or triplicates.

The authors should specify the duration of travel that was considered a travel case. Also, was travel to a foreign country the only consideration, or was travel within the country also considered?

Author´s reply: We do not have reliable information on the duration of travel in most of the diarrhoeal cases. We only considered travel to a foreign country as having a travel history. We have specified that in the methods section of the revised manuscript, lines 132-133.

Did the authors differentiate hospital-acquired diarrhoea from community-acquired diarrhoea? The authors should explain what the hospitalized patients represent.

Author´s reply: In this study we did not differentiate between hospital- and community-acquired diarrhoea. That was not an aim of this study, and we did not have sufficient clinical information to distinguish these two groups. 

The lysis buffer composition or brand name of the reagents used in the study where available should be provided.

Author´s reply: We apologise that this information was not provided. Brand names have been added to the methods section. Lines 157-159.

The authors should explain how they decided on the CT value cutoffs. Did the Multiplex PCR assay provide them?

Author´s reply: The commercial PCR systems provide automatic interpretation of positive versus negative. This information has been added to the methods section. Lines 172-173.

The values described in Line 222-224 should be either ascending or descending.

Author´s reply: We agree and has changed this accordingly. Lines 240-241.

The authors have not described whether any seasonal patterns were observed in the study but have included this as a strength of the study.

Author´s reply: We agree that information about seasonality would be of interest and have added such information in the revised version. Lines 222-226 and 235-236.

It would be interesting to know the authors' thoughts on the discovery of mosaicism of virulence factors in diarrheagenic E. coli pathotypes based on genomic studies.

Author´s reply: This is a very interesting topic, but we do not consider that within the scope of this manuscript. 

The authors should consider reducing the number of discussion points.

Author´s reply: We agree and have attempted to do so.

---

## [Editor Report · Decision Letter 1]

19 Mar 2024

Enteroaggregative Escherichia coli in mid-Norway: A prospective, case control study

PONE-D-23-26927R1

Dear Dr. Haugan,

We’re pleased to inform you that your manuscript has been judged scientifically suitable for publication and will be formally accepted for publication once it meets all outstanding technical requirements.

An invoice for payment will follow shortly after the formal acceptance. To ensure an efficient process, please log into Editorial Manager at Editorial Manager® , click the 'Update My Information' link at the top of the page, and double check that your user information is up-to-date. If you have any billing related questions, please contact our Author Billing department directly at authorbilling@plos.org.

Kind regards,

Furqan Kabir

Academic Editor

PLOS ONE
---

## [Editor Report · Acceptance letter]

27 Mar 2024

PONE-D-23-26927R1 

PLOS ONE

Dear Dr. Haugan, 

I'm pleased to inform you that your manuscript has been deemed suitable for publication in PLOS ONE. Congratulations! Your manuscript is now being handed over to our production team.

Kind regards, 

on behalf of

Dr. Furqan Kabir 

Academic Editor

PLOS ONE